# Uncertain Analysis of Fuzzy Evaluation Model for Water Resources Carrying Capacity: A Case Study in Zanhuang County, North China Plain

**Yinxin Ge [1], Jin Wu [1,*], Dasheng Zhang [2], Ruitao Jia [1] and Haotian Yang [1]**

[1] Faculty of Architecture, Civil and Transportation Engineering, Beijing University of Technology, Beijing 100124, China; geyinxin@emails.bjut.edu.cn (Y.G.); Jiaruitao@emails.bjut.edu.cn (R.J.); yanghaotian723@163.com (H.Y.)
[2] Hebei Institute of Water Science, Shijiazhuang 050051, China; skyzhangdasheng@126.com
[*] Correspondence: WuJin@bjut.edu.cn; Tel.: +86-151-1793-1639

**Abstract:** The scientific and accurate evaluation of water resources carrying capacity has good social, environmental and resource benefits. Reasonable selection of evaluation parameters is the key step to realize efficient and sustainable development of water resources. Taking Zanhuang County in the North China Plain as the research area, this study selected fuzzy comprehensive evaluation models with different weights in the established evaluation index framework to explore the sources of uncertainty affecting the evaluation results of water resources carrying capacity. By using the sensitivity analysis method of index weight, the index with the biggest influence factor on the evaluation result is selected to reduce the uncertainty problems such as index redundancy and small correlation degree. The results show that the correlation and reliable of comprehensive evaluation value obtained by different weight methods is different. The evaluation result obtained by using the analytic hierarchy process is more relevant than the entropy weight method, and it is more consistent with the actual load-bearing situation. The study of sensitivity index shows that water area index is the biggest factor affecting the change of evaluation results, and water resources subsystem and socio-economic subsystem play a dominant role in the whole evaluation framework. The results show that strengthening the data quality control of index assignment and weight method is helpful to reduce the error of water resources carrying capacity evaluation. It can also provide scientific basis for the improvement of fuzzy evaluation model.

**Keywords:** water resources carrying capacity; uncertainty analysis; fuzzy comprehensive evaluation model; weight sensitivity analysis



## 1. Introduction

Water resources are irreplaceable natural resources that not only restrict the sustainable development of society but also play a vital role in social development [1,2]. At present, studying the carrying capacity of water resources is a prerequisite for determining the important development relations between water resources and population, ecology and social economy in the region [3]. It is the necessity to manage the sustainable use of water resources and other related water issues [4]. Water resources and other related water issues must be managed sustainably [5]. Of great significance for promoting the development of the Chinese economy and improving the quality of life is how to effectively realize the balance and sustainable development of water resources, the water environment and the economy [6].

In recent years, the water resources carrying capacity has been discussed more than sustainable development, where the former generally refers to the development and utilization degree of natural water resources [7]. Some studies [8–10] have concluded that they often express similar meanings as indicators such as the sustainable utilization of

water volume, ecological limit of water environment, limit of water resources and shortage degree [11]. Chinese research on the water resources carrying capacity was first proposed by the Xinjiang Water Resources Soft Science Research Group [12] and constituted a breakthrough in the field of water resources. In the larger theoretical context of sustainable development and water management, the most representative definition is to coordinate the reasonable scale of ecological health and sustainable development resources under certain social conditions of economic, environmental and technological development [13]. Water resource carrying capacity is defined as "the size of population and economy scale that a region's water resources can carry, which had necessary requirements for ecological environmental protection and had certain technical level and social and economic development level in a certain historical stage". Since the water resources carrying capacity involves water resources system, ecological environment system and social economic system under different regional and natural conditions, the interaction among multiple systems will further amplify the complexity and uncertainty [14]. Therefore, strengthening the study of uncertainty of water resources carrying capacity is conducive to improving the reliability of evaluation results. Among the comprehensive dynamic evaluation models, the fuzzy comprehensive evaluation model [15] is widely used. In addition, fuzzy comprehensive evaluation manages fuzzy evaluation variables through accurate mathematical methods, which can provide a more scientific and practical quantitative evaluation of hidden and fuzzy concepts. At the same time, the model can be used to verify whether the evaluation index weight and other related uncertainty issues have a great impact on the evaluation results [16]. Therefore, the fuzzy evaluation model is selected as an example to study the uncertainty of the evaluation results of the water resources carrying capacity [17,18]. From the perspective of weight data and indicator assignment, the uncertainty of water resources carrying capacity is seldom considered in the application of the fuzzy evaluation model, which makes the results of water resources carrying capacity limited in practical applications [11,15,19]. Therefore, the uncertainty study of water resources carrying capacity model will improve the accuracy of evaluation. At the same time, the research on the uncertainty of water resources carrying capacity also provides a direction for the improvement of the model [20–22].

Based on the consideration of these uncertain factors, this paper studies the fuzzy comprehensive evaluation model by comparing the weight method determined by the analytic hierarchy process and the entropy weight method [23,24] under certain technical outline standards. Correlation analysis was used to solve the uncertainty among weights, the indexes with high sensitivity coefficient and great influence on evaluation results were screened out by calculating the influence of indexes through sensitivity analysis [25]. This study provides reliable scientific basis for reducing the uncertainty of the results of the fuzzy evaluation model and improving the efficiency of water resources utilization [26,27]. Taking Zanhuang County in the North China Plain as an example, the evaluation of water resources carrying capacity is carried out with certain characteristic parameters [28,29], which can provide reference for water resources managers in the North China Plain.

## 2. Materials and Methods

### 2.1. Study Area

The study area (37°26′ to 37°46′ N, 114°20′ to 114°31′ E) is located in Shijiazhuang city in southwestern Hebei Province (Figure 1b). The district's east–west length is 44.8 km, and its north–south width is 37 km, resulting in a total area of 1210 km$^2$ (Figure 1c). It borders the surrounding counties of Shijiazhuang City and Xingtai City inside, and Shanxi Province outside. The county is located in a warm, semi-humid monsoon continental climate. The temperature difference between seasons is large, with an annual average temperature of 13.6 °C. The average annual precipitation and evaporation are 508.9 mm and 1885 mm, respectively, and precipitation mainly occurs from July to September. Regarding its topography, the region is located at the eastern foot of the Tai-hang Mountains. The landform features of the region are mainly composed of mountains and hills. On the

whole, the mountain trend is higher in the west and lower in the east. The abundant basin-scale crossing of water resources from the Ziya River System and the artificial canal South-to-North Water Diversion Project in the study area can meet 68% of the county's annual water consumption. The county has 11 towns under its jurisdiction.

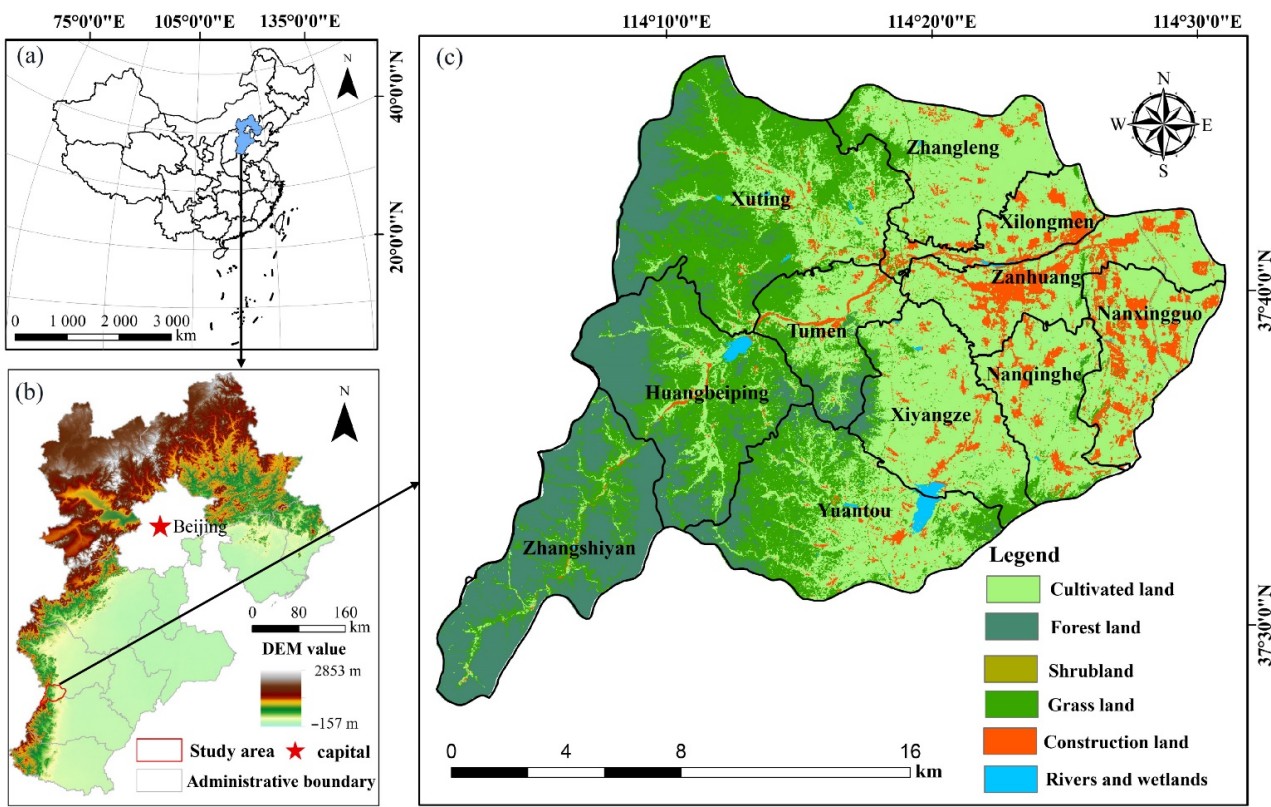

**Figure 1.** Maps showing (**a**) the location of Hebei in China; (**b**) the location and elevation of study area in Hebei; (**c**) the administrative boundaries in the study area combined with its land use.

*2.2. Methods*

2.2.1. Fuzzy Comprehensive Evaluation Model

As water resources carrying capacity evaluation index standards and evaluation system boundaries are usually uncertain and fuzzy, fuzzy comprehensive evaluation models can improve the objectivity and accuracy of evaluation results and more comprehensively reflect the situation of regional water resources more comprehensively [30]. This paper adopts this model and its basic principle is as follows: to establish evaluation index set $U = (u_1, u_2, \ldots, u_m)$ and the comment set $V = (v_1, v_2, \ldots, v_n)$, the results of fuzzy comprehensive evaluation are as follows:

$$C = (c_1, c_2, \ldots, c_m) = W \cdot R \tag{1}$$

where W is a fuzzy subset on U, $W = (w_1, w_2, \ldots, w_n)$, $0 \leq w_i \leq 1$ ($\sum_{i=1}^{n} w_i = 1$); $w_i$ is the membership degree of U to W (the weight value of this indicator), which represents the extent to which a single element $u_i$ plays a role in the evaluation factor; "·" is a fuzzy operator which ordinary matrix algorithm is adopted; C is a fuzzy subset of V, $C = (c_1, c_2, \ldots, c_m)$, $0 \leq c_j \leq 1$ ($\sum_{j=1}^{m} c_j = 1$); $c_j$ is the membership degree of V to C, which represents the result of comprehensive evaluation. The membership (evaluation) matrix is as follows:

$$R = \begin{bmatrix} r_{11} & \cdots & r_{1n} \\ \ldots & \ddots & \ldots \\ r_{m1} & \cdots & r_{mn} \end{bmatrix} \tag{2}$$

In the formula, $r_{ij}$ is the membership degree of evaluation $u_i$ to $v_j$, and $R_i = (r_{i1}, r_{i2}, \ldots, r_{im})$ is the result of single factor for evaluation of $u_i$.

The actual value of the evaluation index $u_i$ is compared with the classification interval, the membership degree of the corresponding level $v_j$, which is named the value of $r_{ij}$, can be calculated. In order to eliminate the level skip phenomenon in which the value of the evaluation grade changes in a small range at the end of the section, the membership function can be smoothly transitioned between each level, and the fuzzy processing is performed. When calculating the membership degree matrix R, $r_i^{(t)}$ (t = 1, 2, 3, 4, 5,), which means the membership of the t-th level, $r_i$ is the actual value of the index, and $x_{max}^{(t)}$ and $x_{min}^{(t)}$ refer to the upper limit and lower limit of the t-th evaluation level, respectively. The algorithm of the membership matrix R is not elaborated in detail [31].

After obtaining fuzzy comprehensive evaluation matrix C of each town, each grade of the evaluation index reflects different situations of water resources carrying capacity [32]. A value between 0 and 1 is assigned to each grade for quantification, and the larger the value is, the stronger the water resources carrying capacity is. Generally speaking, the evaluation indexes were divided into five levels, and the comment set $V = (v_1, v_2, \ldots, v_5)$. The $v_1$ level indicates that the carrying capacity of water resources is in a best state, and the coordinated development of water resources with economy, society and ecology is in a state of sustainable utilization. The $v_2$ level indicates that the carrying capacity of water resources is in a good state, and the water resource is sufficient to support the local economic and social development level. The $v_3$ level indicates that the carrying capacity of water resources is in the general state and there is no obvious regional water shortage problem. The $v_4$ level indicates that the carrying capacity of water resources is in a poor state, but it can basically meet the water demand of various industries. The $v_5$ level indicates that the carrying capacity of water resources is in a very poor state and the contradiction of water resources is prominent. Take $\alpha_1 = 0.1$, $\alpha_2 = 0.3$, $\alpha_3 = 0.5$, $\alpha_4 = 0.7$ and $\alpha_5 = 0.9$ for levels $v_1$, $v_2$, $v_3$, $v_4$ and $v_5$, respectively. The scoring value of each grade and the final comprehensive evaluation value of water resources carrying capacity are calculated according to the formula below.

$$\theta = \frac{\sum_{t=1}^{5} b_t^k \alpha_t}{\sum_{t=1}^{5} b_t^k} \tag{3}$$

where $\theta$ is the comprehensive evaluation value of water resources carrying capacity based on the comprehensive evaluation result matrix C; $b_t^k$ is the value of the membership degree of each evaluation index; k is the coefficient set when the dominant role needs to be highlighted, usually k = 1.

### 2.2.2. Index Weight Calculation Methods

The weight calculation methods can be divided into subjective methods and objective methods [33,34], including the binomial coefficient method and the analytic hierarchy process. The subjective method research is relatively mature, with strong subjective arbitrariness, and more dependence on the thinking of the decision analyst. However, the principal component analysis, entropy and other objective methods use decision matrices, which have a strong mathematical theoretical basis to determine weights based on relationships between the original data. Since many factors are involved in the water resources carrying capacity, and different factors have different effects on it, the actual situation of the study area should be considered when assigning weights [35]. First, the relationship between the indicators was clarified, and the corresponding index system was established. The index system was divided into three levels: target level, criterion level and index level [36]. In this paper, the analytic hierarchy process and the entropy weight method were used to discuss and study this respectively, and an appropriate weight method was found.

Analytic Hierarchy Process

The analytic hierarchy process (AHP) is a systematic method of making decisions by means of qualitative indicators and fuzzy quantification [30]. According to the nature of the problem and the overall goal to be achieved, it deconstructs the problem into different constituent factors. The AHP combines the factors at different levels according to their interrelationship and the affiliation relationship, forming a multilevel analysis structure model. Thus, ultimately, the problem is attributed to the determination of the importance of the lowest level (plans, measures, etc. for decision making) relative to the highest level (the overall goal) or the arrangement of the relative order of superiority and inferiority. The main steps of the analytic hierarchy process are as follows:

(1) Establish the hierarchical structure model;
(2) Construct judgment matrix by comparing paired indexes;
(3) Calculate the maximum eigenvalue and eigenvector of the judgment matrix, and carry out the consistency test;
(4) Calculate the weight of each evaluation index.

Entropy Weight Method

The entropy weight method is used to determine the weight of each evaluation index. Generally, when the information entropy of an index is smaller, the information provided and the index weight is greater, and vice versa [37]. The main calculation steps are as follows:

The original evaluation index matrix B was obtained according to the membership:

$$B = \begin{bmatrix} b_{11} & \cdots & b_{1n} \\ \dots & \ddots & \dots \\ b_{m1} & \cdots & b_{mn} \end{bmatrix} \tag{4}$$

where $b_{ij}$ is the original value of the i-th index in the j-th year. The normalized matrix A is obtained by eliminating the dimensional effect. The positive and negative indicators are treated as follows:
Positive indicators:

$$a_{ij} = \frac{b_{ij} - \min(b_{ij})}{\max(b_{ij}) - \min(b_{ij})} \tag{5}$$

Negative indicators:

$$a_{ij} = \frac{\max(b_{ij}) - b_{ij}}{\max(b_{ij}) - \min(b_{ij})} \tag{6}$$

Normalization matrix:

$$A = \begin{bmatrix} a_{11} & \cdots & a_{1n} \\ \dots & \ddots & \dots \\ a_{m1} & \cdots & a_{mn} \end{bmatrix} \tag{7}$$

Calculation of objective weight through the entropy weight method:

$$w_i = \frac{1 - e_i}{m - \sum_{i=1}^{m} e_i} \tag{8}$$

Information entropy:

$$e_i = -\frac{1}{\ln n} \sum_{j=1}^{n} p_{ij} \ln p_{ij}, \, p_{ij} = \frac{a_{ij}}{\sum_{i=1}^{n} a_{ij}} \tag{9}$$

2.2.3. Calculation Method of Weight Sensitivity

In the evaluation of water resources carrying capacity, the weight of each index in the evaluation index system can be obtained with the help of fuzzy comprehensive evaluation

model, but it cannot judge which index has a high impact on the evaluation. Sensitivity analysis can explain the influence of which index weight by changing the value of relevant variables. It is an essential basic step in the process of multi-criteria decision making, because it is directly related to the accuracy and reliability of decision-making results [38].

This paper adopted the single-factor division method [39] to test the sensitivity of the index weight, which can be shown by removing a certain variable weight. The weights of the other variables were equally distributed to remove the variable weight value and to maintain the total weight value and the value of 1. The changing situation reflects the trend and regularity of the influence of single-factor weight changes on the water resources carrying capacity and then removes the weights of other indicators individually, calculates their respective sensitivities and evaluates the impact of the uncertainty of the weight of each indicator on the research results variation. If removing this index weight does not have a great impact on the score result, then the comprehensive evaluation of the water resources carrying capacity is insensitive to this index weight, and vice versa. The calculation of this method is shown in Formulas (10) and (11).

$$\mathrm{RMSEC} = \sqrt{\frac{\sum_{i=1}^{n}\left(\frac{Y - Y_i}{Y}\right)^2}{n}} \tag{10}$$

$$\mathrm{TF} = \sum_{i=1}^{k} F_{\mathrm{RMSEC}} \tag{11}$$

where RMSEC is the rate of change of root mean square error (the sensitivity index); n is the number of index weight variables; Y is the comprehensive evaluation value in the original fuzzy comprehensive evaluation results; $Y_i$ is the comprehensive evaluation value after changing the weight index variable; TF is the total sensitivity; K is the number of comprehensive evaluation results; $F_{\mathrm{RMSEC}}$ is the change rate of the root mean square error of the comprehensive evaluation results, that is, the sensitivity of each weight variable in each comprehensive evaluation after changing.

Firstly, the evaluation of water resources carrying capacity is calculated under the basic framework of the fuzzy comprehensive evaluation model, and the framework of the evaluation system of water resources carrying capacity is established. Then, according to the original data, it is processed and indexed according to the grading standards. Secondly, the reliability analysis of the comprehensive evaluation value obtained by different weights was compared by using correlation analysis method. The error contrast of mathematical objectivity was analyzed under a certain research report standard. The evaluation index value of water resources carrying capacity under the selected weight results was calculated. Thirdly, the sensitivity analysis method of single variable removal is used to calculate the sensitivity index of these evaluation values to screen out the index that has the greatest influence on the evaluation results. For this research, the uncertainty methods provide ideas for the systematic model study of water resource carrying capacity evaluation, Besides this, it also provides direction for the improvement of the model.

### 3. Results and Discussion

#### 3.1. Construction of the Evaluation Index System and Classification Standard

The selection of indicators is directly related to the accuracy and authenticity of the evaluation results of water resources carrying capacity, so the selection of evaluation indicators should follow the principles of science, integrity, hierarchy, dynamic and operability [40,41]. To accurately reflect the status of the water resources carrying capacity in this region, indices were selected from two aspects. On the one hand, when selecting indicators, we mainly considered our available data (The Shijiazhuang Water Resources Bulletin, Statistical Yearbook of Zanhuang) [42,43] and referred to indicators in other literature [39–41,44]; on the other hand, according to the actual situation and characteristics of water resources in Hebei Province, combined with experts' opinions, the indexes of the

study area were selected comprehensively [45]. Finally, the total system was divided into four subsystems: water resources, water environment, water ecology and social economy. A total of 13 evaluation indexes were selected to construct the evaluation index system of water resources carrying capacity in Zanhuang County (Table 1). If the index value increases indefinitely and approaches the $V_1$ standard with better carrying capacity, it is a positive index. On the contrary, as the index value increases infinitely and approaches the $V_5$ standard with poor carrying capacity, it is a negative index. The definition and criterion of each indicator in the index layer are shown in Table 2.

*3.2. Comparison of Weight Results between the Analytic Hierarchy Process and the Entropy Weight Method*

Different weights should be assigned to various evaluation indices because of their different effects on water resources carrying capacity. Weight is a very important factor in fuzzy comprehensive evaluation, and its accuracy directly affects the rationality of the evaluation results. The data sources used in this paper are mainly from the Shijiazhuang Water Resources Bulletin (2017) and Statistical Yearbook of Zanhuang in 2017. The above two calculation methods can be used to obtain the weights of the four subsystems of the analytic hierarchy process (detailed calculation process can be found in the supplementary materials section in Tables S1 and S2), which are respectively $W_A = 0.46$, $W_B = 0.14$, $W_C = 0.13$, $W_D = 0.27$; the weight of the four subsystems of the entropy weight method is $W_A' = 0.20$, $W_B' = 0.38$, $W_C' = 0.16$, $W_D' = 0.26$. The final weights calculated by each index layer are shown in Figure 2.

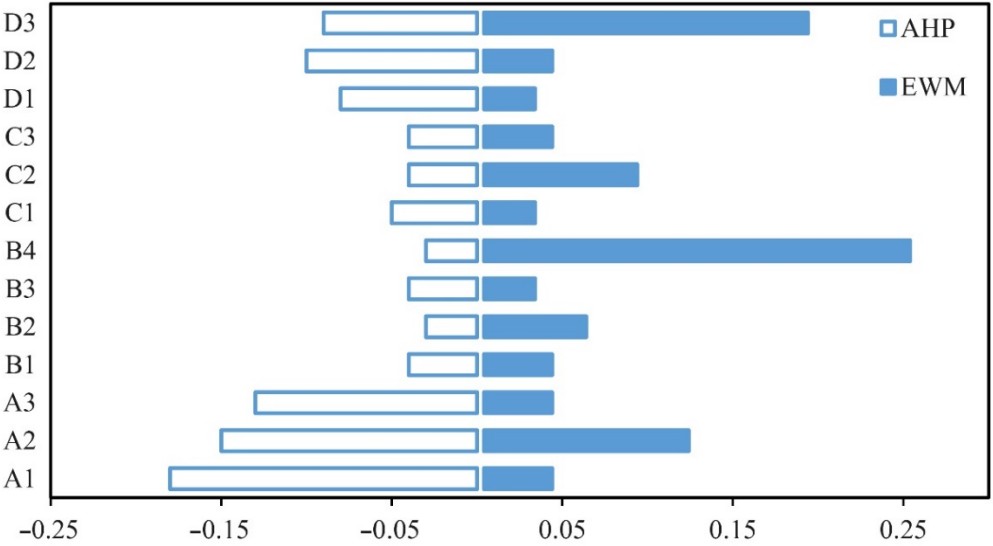

**Figure 2.** Comparison of weight results calculated by AHP and EWM.

From Figure 2, the five indexes ranked from high to low of the analytic hierarchy process are water resources development and utilization rate, water consumption per unit of GDP, water area index, per capita GDP and domestic water quota, with weights of 0.18, 0.15, 0.13, 0.10 and 0.09, respectively, accounting for 65% of the total contribution rate, including the water resources subsystem and the socio-economic subsystem, which should be the focus of improving the water resources carrying capacity of study area. The other indexes have relatively little influence on the evaluation results. This result verifies that the analytic hierarchy process comprehensively considers the coupling effect between multiple criteria and multiple indicators, and focuses on the importance of identifying effective indicators.

**Table 1.** Comprehensive evaluation index and classification standard of water resources carrying capacity in Zanhuang County.

| Target Layer | Criterion Layer | Index Layer | Index Type | Code | Unit | Classification Standard | | | | |
|---|---|---|---|---|---|---|---|---|---|---|
| | | | | | | V$_1$ | V$_2$ | V$_3$ | V$_4$ | V$_5$ |
| Comprehensive water resources carrying capacity evaluation index system | Water Resources Subsystem (A) | Water resources development and utilization rate | Negative | A$_1$ | % | <15 | 15–20 | 20–35 | 35–60 | >60 |
| | | Water consumption per unit of GDP | Negative | A$_2$ | m$^3$/10$^4$ yuan | <50 | 50–75 | 75–80 | 80–100 | >100 |
| | | Water area index | Positive | A$_3$ | % | >5 | 4–5 | 3–4 | 2–3 | <2 |
| | Water Environment Subsystem (B) | Water environmental quality index | Positive | B$_1$ | % | >90 | 80–90 | 70–80 | 60–70 | <60 |
| | | Industrial wastewater discharge index | Negative | B$_2$ | % | <10 | 10–20 | 20–40 | 40–50 | >50 |
| | | Fertilizer intensity index | Negative | B$_3$ | kg/hm$^2$ | <100 | 100–150 | 150–200 | 200–250 | >250 |
| | | Urban sewage discharge index | Negative | B$_4$ | % | <10 | 10–20 | 20–40 | 40–50 | >50 |
| | Water Ecological Subsystem (C) | The vegetation coverage rate of coastal zone | Negative | C$_1$ | % | <20 | 20–30 | 30–40 | 40–60 | >60 |
| | | Ecological base flow guarantee rate | Positive | C$_2$ | % | >60 | 40–60 | 30–40 | 20–30 | <20 |
| | | River network density index | Positive | C$_3$ | 1/km | >0.8 | 0.6–0.8 | 0.4–0.6 | 0.2–0.4 | <0.2 |
| | Socioeconomic Sub-system (D) | Population density | Negative | D$_1$ | 1/km$^2$ | <300 | 300–500 | 500–700 | 700–900 | >900 |
| | | Per capital GDP | Positive | D$_2$ | 10$^4$ yuan | >7.5 | 6–7.5 | 4.5–6 | 3–4.5 | <3 |
| | | Domestic water quota | Positive | D$_3$ | liter/day | >130 | 110–130 | 90–110 | 70–90 | <70 |

**Table 2.** Definition and criterion of each indicator in the index layer.

| Indicator | Definition | Criterion |
|---|---|---|
| | Water Resources Subsystem | |
| Water resources development and utilization rate | Regional water consumption/regional water resources | The Shijiazhuang Water Resources Bulletin |
| Water consumption per unit of GDP | Regional water consumption/total regional GDP | The Shijiazhuang Water Resources Bulletin and Statistical Yearbook of Zanhuang |
| Water area index | Area of water area/ the total area | Statistical Yearbook of Zanhuang and Google Satellite Map |
| | Water Environment Subsystem | |
| Water environmental quality index | The rate of water quality discharge up to standard | Environmental monitoring Reports |
| Industrial wastewater discharge index | Regional industrial water discharge/total wastewater discharge | The Shijiazhuang Water Resources Bulletin and Environmental monitoring Reports |
| Fertilizer intensity index | Total amount of fertilizer applied (discounted)/cultivated area of evaluation area | Statistical Yearbook of Zanhuang |
| Urban sewage discharge index | Regional urban sewage discharge/total wastewater discharge | The Shijiazhuang Water Resources Bulletin and Environmental monitoring Reports |
| | Water Ecological Subsystem | |
| The vegetation coverage rate of coastal zone | Length of plant cover/length of shoreline | Statistical Yearbook of Zanhuang and Google Satellite Map |
| Ecological base flow guarantee rate | Average monthly actual flow/minimum ecological flow | Rain station monitoring reports |
| River network density index | River length/watershed area | Statistical Yearbook of Zanhuang and Google Satellite Map |
| | Socioeconomic Subsystem | |
| Population density | Regional population/regional administrative area | Statistical Yearbook of Zanhuang |
| Per capital GDP | Regional GDP/regional population | Statistical Yearbook of Zanhuang |
| Domestic water quota | Domestic water consumption/ (regional population· days) | The Shijiazhuang Water Resources Bulletin and Statistical Yearbook of Zanhuang |

The weights of the first five indicators of the entropy method are urban sewage discharge index (0.25), domestic water quota (0.19), water consumption per unit of GDP (0.12), ecological base flow guarantee rate (0.09) and industrial wastewater discharge index (0.06). The first five indicators calculated by this method cover the four sub-systems of the entire criterion, namely, water resources, water environment, water ecology and social economy. First, according to the formula there is no horizontal comparison between the indicators in the calculation process, in that there is no distinction between primary and common indicators. Second, the weight value is too mathematically objective, which covers all the subsystems in the five indicators ranked from high to low. Therefore, the coupling and correlation of the main influencing indicators are often ignored, and it is limited in practical applications. The mutual influence between indicators cannot be ignored for accurate evaluation.

### 3.3. Comparative Study of Fuzzy Comprehensive Evaluation

It can be seen from Figure 2 that there is a big difference between the weight results of the analytic hierarchy process and the entropy weight method. Tables 3 and 4 are the comprehensive evaluation results of the analytic hierarchy method and the entropy weight

method, respectively. Table 5 presents the correlation analysis, in which IBM SPSS Statistics 25 software was used to illustrate this more clearly.

**Table 3.** Comprehensive evaluation results of water resources carrying capacity by the analytic hierarchy process in Zanhuang County.

| Sites | $V_1$ | $V_2$ | $V_3$ | $V_4$ | $V_5$ | Comprehensive Evaluation Value θ | Theta Ranked from High to Low |
|---|---|---|---|---|---|---|---|
| Zanhuang | 0.2500 | 0.0026 | 0.0879 | 0.1395 | 0.5000 | 0.6174 | 3 |
| Xilongmen | 0.1700 | 0.1813 | 0.2056 | 0.2107 | 0.3124 | 0.6029 | 6 |
| Nanxingguo | 0.3900 | 0.1811 | 0.1833 | 0.0956 | 0.2300 | 0.4589 | 10 |
| Nanqinghe | 0.3277 | 0.1838 | 0.1015 | 0.0971 | 0.2900 | 0.4676 | 9 |
| Yuantou | 0.1300 | 0.2708 | 0.2182 | 0.1519 | 0.3300 | 0.6067 | 5 |
| Xiyangze | 0.3550 | 0.1186 | 0.0364 | 0.0000 | 0.4900 | 0.5303 | 8 |
| Tumen | 0.1800 | 0.0950 | 0.1528 | 0.1622 | 0.4900 | 0.6774 | 1 |
| Huangbeiping | 0.1000 | 0.1932 | 0.2202 | 0.0166 | 0.4700 | 0.6127 | 4 |
| Zhangshiyan | 0.2433 | 0.0636 | 0.0766 | 0.0470 | 0.5700 | 0.6276 | 2 |
| Xuting | 0.2600 | 0.1641 | 0.1559 | 0.0400 | 0.4600 | 0.5952 | 7 |
| Zhangleng | 0.3750 | 0.1232 | 0.1654 | 0.0865 | 0.2500 | 0.4427 | 11 |

**Table 4.** Comprehensive evaluation results of water resources carrying capacity by the entropy weight method in Zanhuang County.

| Scheme 1. | $V_1$ | $V_2$ | $V_3$ | $V_4$ | $V_5$ | Comprehensive Evaluation Value θ | Theta Ranked from High to Low |
|---|---|---|---|---|---|---|---|
| Zanhuang | 0.5434 | 0.0021 | 0.0733 | 0.1049 | 0.2582 | 0.3974 | 9 |
| Xilongmen | 0.5594 | 0.1436 | 0.1492 | 0.1476 | 0.0729 | 0.3426 | 10 |
| Nanxingguo | 0.3701 | 0.1275 | 0.0973 | 0.1821 | 0.2229 | 0.4520 | 7 |
| Nanqinghe | 0.2906 | 0.3073 | 0.2385 | 0.0682 | 0.1653 | 0.4370 | 8 |
| Yuantou | 0.4672 | 0.0549 | 0.0271 | 0.0000 | 0.4507 | 0.4824 | 6 |
| Xiyangze | 0.3245 | 0.0717 | 0.1416 | 0.0839 | 0.4507 | 0.5891 | 2 |
| Tumen | 0.0370 | 0.2726 | 0.2963 | 0.0055 | 0.3885 | 0.5872 | 3 |
| Huangbeiping | 0.1093 | 0.1729 | 0.2006 | 0.0409 | 0.4771 | 0.6211 | 1 |
| Zhangshiyan | 0.3339 | 0.1360 | 0.1247 | 0.0362 | 0.4415 | 0.5593 | 4 |
| Xuting | 0.5464 | 0.1255 | 0.1535 | 0.0358 | 0.1389 | 0.3190 | 11 |
| Zhangleng | 0.3775 | 0.1435 | 0.1193 | 0.2333 | 0.1989 | 0.4827 | 5 |

**Table 5.** Correlation analysis of the membership values of two weighting methods and evaluation values.

| Weighting Methods | Evaluation Level | Mean Value | Standard Deviation | $V_1$ | $V_2$ | $V_3$ | $V_4$ | $V_5$ | θ |
|---|---|---|---|---|---|---|---|---|---|
| AHP | $V_1$ | 0.2528 | 0.1003 | 1 | | | | | |
| | $V_2$ | 0.1434 | 0.0729 | −0.251 | 1 | | | | |
| | $V_3$ | 0.1458 | 0.0620 | −0.512 | −0.679 * | 1 | | | |
| | $V_4$ | 0.0952 | 0.0661 | −0.303 | 0.073 | 0.391 | 1 | | |
| | $V_5$ | 0.3993 | 0.1182 | −0.368 | −0.575 | −0.488 | −0.361 | 1 | |
| | θ | 0.5672 | 0.0791 | −0.815 ** | 0.679 * | 0.871 ** | 0.235 | 0.737 ** | 1 |
| EWM | $V_1$ | 0.3599 | 0.1715 | 1 | | | | | |
| | $V_2$ | 0.1416 | 0.0883 | −0.638 * | 1 | | | | |
| | $V_3$ | 0.1474 | 0.0756 | −0.717 * | 0.871 ** | 1 | | | |
| | $V_4$ | 0.0853 | 0.0750 | 0.316 | −0.148 | −0.266 | 1 | | |
| | $V_5$ | 0.2969 | 0.1477 | −0.561 | −0.159 | −0.031 | −0.560 | 1 | |
| | θ | 0.4791 | 0.1019 | −0.833 ** | 0.192 | 0.299 | −0.299 | 0.869 ** | 1 |

Note: *: $p < 0.05$，significant correlation; **: $p < 0.01$, extremely significant correlation.

The reason for this result is that the entropy weight method has some shortcomings. First, the number of indicators selected in this evaluation is larger than the number of objects to be evaluated, resulting in deviations. Second, the entropy weight method ignores the importance of the index and excessively relies on an objective weight assignment, which causes the dimension of the evaluation index to not be reduced and the subjective intention

of the decision-maker to be ignored [32,46]. At the same time, because each township is independent and different, the weight of each index layer should be different. This factor was not well represented in this method, and many similar weight results were obtained during calculation. The analytic hierarchy process comprehensively considers the coupling effect between multiple criteria and multiple indicators according to the intention of the decision-makers and the actual local situation. Moreover, the method more effectively identifies the importance of the main influencing indicators [47].

In Table 5, we have found that among the five membership values obtained by the analytic hierarchy process, four membership values ($v_1$, $v_2$, $v_3$, $v_5$) are significantly correlated with the evaluation value theta, the contribution rate of correlation accounted for 80%. Among the five membership values obtained by the entropy weight method, only two membership values ($v_1$, $v_5$) are significantly correlated with theta and the contribution rate of correlation accounted for 40%. It indicates that the correlation contribution rate of evaluation results obtained by using the analytic hierarchy process is higher than the entropy weight method. The reliability of the analytic hierarchy process is higher. In addition, the average evaluation value theta of the analytic hierarchy process (0.5672) obtained is higher than the entropy weight method (0.4791), and the standard deviation of the former (0.0791) is lower than the latter (0.1019). It also indicates that the error result is smaller. Besides this, the evaluation value obtained by the analytic hierarchy process is also more satisfied with the degree of non-overloading of the evaluation results in the evaluation report of Carrying capacity in Hebei Province. In summary, it shows that the analytic hierarchy process in this study is more suitable for the actual situation.

According to the existing research results, combined with the water resources conditions, ecological environment characteristics and social development of Hebei Province, the comprehensive grading standards are obtained (Table 6).

**Table 6.** Classification criteria of comprehensive score values of water resources carrying capacity.

| Evaluation Results | 0–0.25 | 0.25–0.50 | 0.50–0.75 | 0.75–1.00 |
|---|---|---|---|---|
| Bearing level | Unbearable | General bearing | Good bearing | Ideal bearing |

*3.4. Analysis of Evaluation Results*

In general, the scores of the 11 towns and villages in the comprehensive evaluation value determined by the analytic hierarchy process were between 0.4 and 0.7. This result indicates that the water resources carrying capacity of the region is in a general and a good bearing capacity. Water resources remain to be exploited, so the local economy can maintain its development. Domestic and ecological water use is in a relatively balanced state, and there will not be a serious water shortage in the near term.

According to the comprehensive score, Tumen, Zhangshiyan, Zanhuang, Huangbeiping, Yuantou, Xilongmen, Xuting, Xiyangze, Nanqinghe, Nanxingguo and Zhangleng townships are ranked from high to low. In calculating the comprehensive evaluation value, the weight value of each subsystem is different. When the scores are the same, the score value of the subsystem with a larger weight value has a greater impact than does the evaluation result of the subsystem with a smaller weight value [48]. In order to better show the contribution of each subsystem's score to the comprehensive evaluation value, this paper calculates the accumulation of each subsystem in the comprehensive evaluation value of each township to explore the impact of each subsystem. The results are shown in Figure 3. The depth of the color orange on the map is used to reflect the final evaluation value of each region. The darker the color, the higher the evaluation value, and the lighter the color, the smaller the evaluation value.

Because natural water resources and the distribution of local industry are limited, various regions have different water resource conditions, ecological environments, and social and economic development patterns [49]. Therefore, the water resources carrying capacity of 11 towns shows a certain degree of spatial differentiation. To reflect the spatial

divergence of various regions at the subsystem level more clearly, this paper conducted independent fuzzy comprehensive calculations on the water resources, water environment, water ecological and socioeconomic subsystems. After obtaining the comprehensive score value of each system, the results obtained were based on the Kriging interpolation method to carry out optimal unbiased interpolation research on the data results to reduce the uncertainty of the evaluation results. This process was mainly due to the consideration of spatial correlation and independence, which makes the results more reliable. The results are shown in Figure 4.

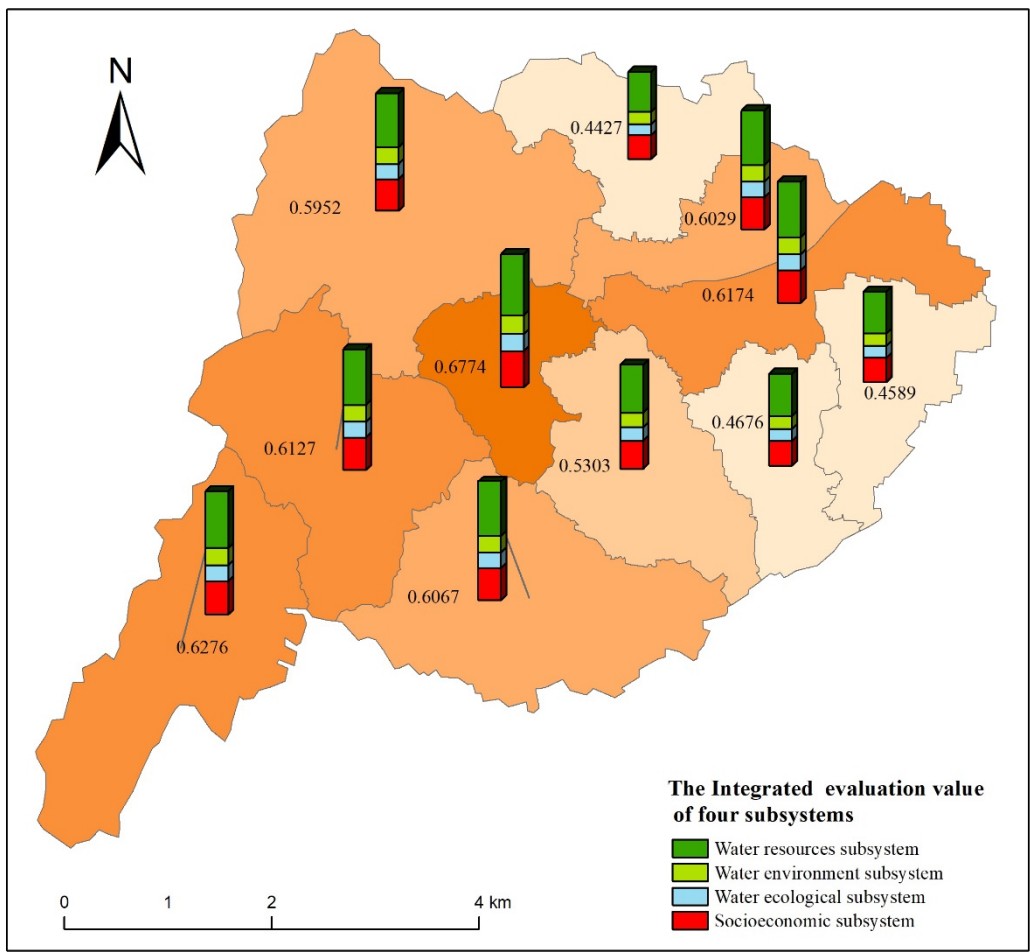

**Figure 3.** Comprehensive scoring values and accumulation of water resources carrying capacity in Zanhuang County.

Figure 4 shows that the spatial distribution of the four subsystems to the water resources carrying capacity was different because of the different index weights. For (a), the high scoring value was concentrated in the eastern part of the study area, which is far from the mountainous area. However, the other parts of the study area, especially Tumen, mainly consumed a large amount of water for irrigation, leading to a high utilization rate of water resources. Meanwhile, a small number of river systems pass through this area, so the water area index was low. In contrast, the score was higher in the central and western regions due to the distant distribution of industrial and urban centers for (b). The eastern part of the county had a lower score due to the discharge of domestic sewage and industrial wastewater. The difference was significant between agricultural areas and wetland rivers in the (c) regional distribution. The high and low values were more obvious when reflecting the high vegetation coverage and the construction of new river channels. The low value of the (d) region was mostly distributed in the middle and northeastern parts of the study area, which have a high per capita GDP, but the domestic water quota and population

density were much higher than they were in other districts. The contribution of high scores in other regions may be influenced by the urban–rural integration.

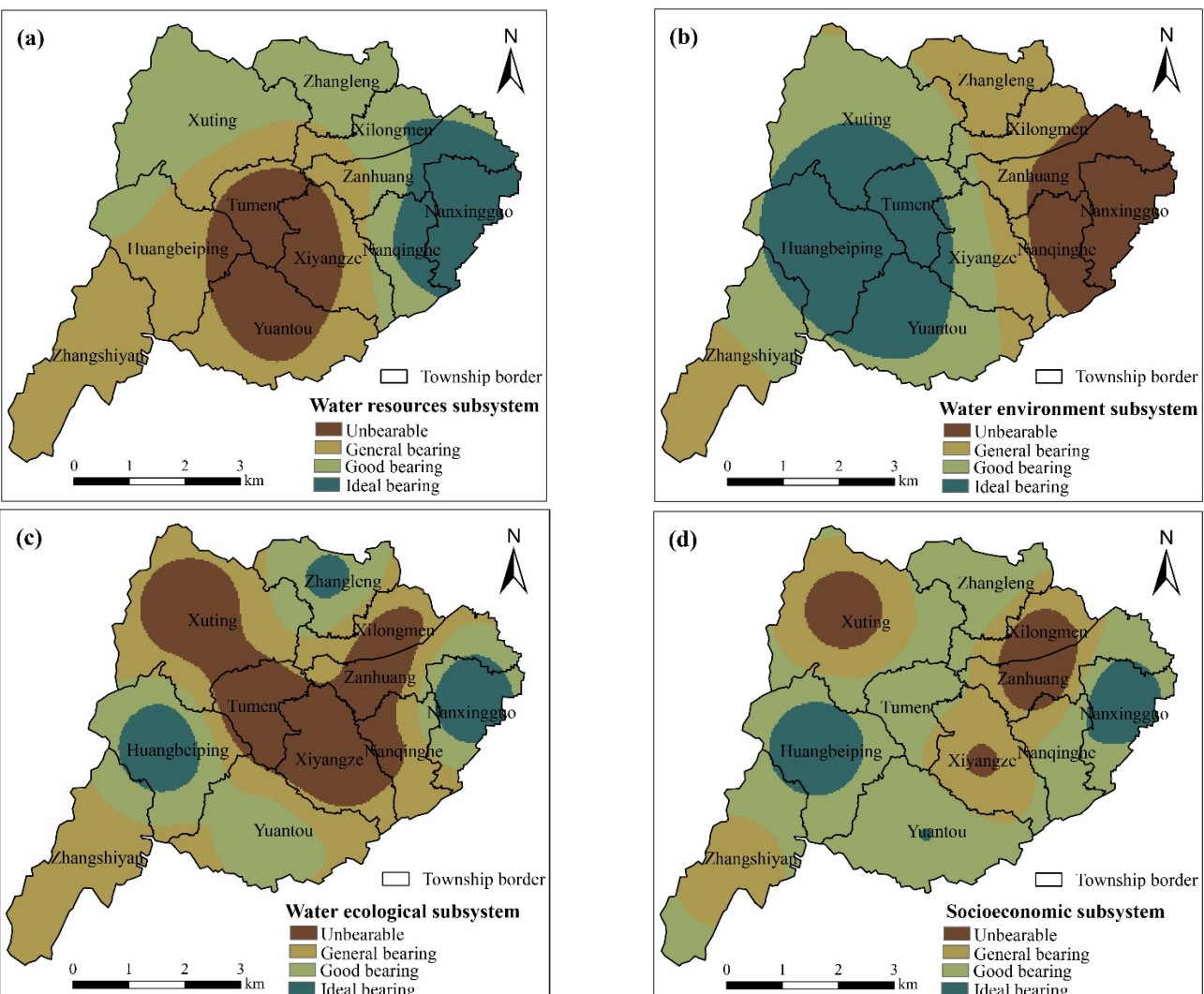

**Figure 4.** Maps with the spatial distribution of four subsystems' classification: (**a**) Water resources subsystem quality classification from scoring values; (**b**) Water environment subsystem quality classification from scoring values; (**c**) Water ecological subsystem quality classification from scoring values; (**d**) Socioeconomic subsystem quality classification from scoring values.

### 3.5. Sensitivity Analysis Results of Weight

This paper used a fuzzy comprehensive evaluation model of weight determined by the analytic hierarchy process. It determined the sensitivity distribution of the weight change of the index variable. Figure 5 compares the comprehensive evaluation value of each township with and without one indicator weight removed. "BASE" represents the result of the comprehensive evaluation value calculated under the condition of all indices, while "-xx" represents it after removing the weight of this evaluation index. Figure 5 indicates that when certain indicators were removed separately, there were many differences in the results of some comprehensive evaluation values. Therefore, the weights of these indicators are highly sensitive. To further study the sensitivity of the specific index weight to the comprehensive evaluation value of the water resources carrying capacity, the RMSEC of each township after the weight change of each index was quantitatively calculated and added to obtain the total sensitivity, TF. The calculation results are shown in Figure 6.

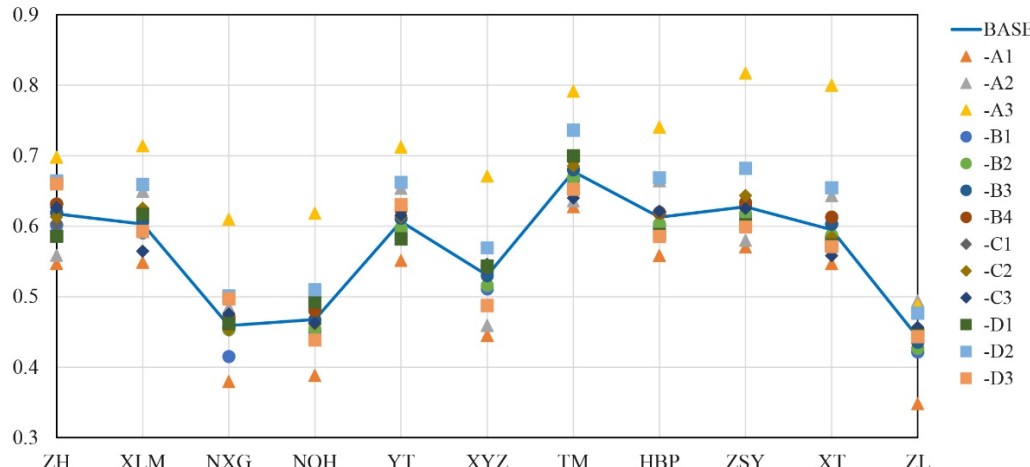

**Figure 5.** Comparison of the comprehensive score values of each township between original and single index weight removed. (The meaning of the abscissa in the figure is to use the first letters of the names of the locations to represent the towns. The indexes like "A1, A2…D3" express the meanings of index layer in Table 1 in turn.)

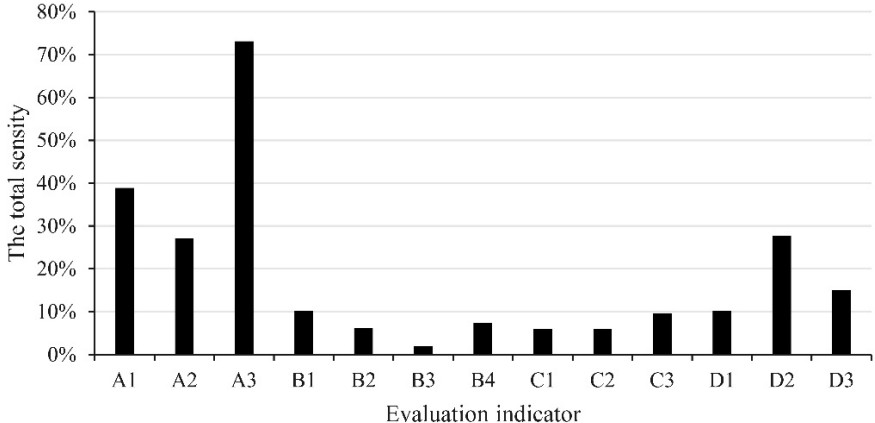

**Figure 6.** Distribution of total sensitivity of evaluation indexes of water resource carrying capacity. (The indexes like "A1, A2…D3" express the meanings of index layer in Table 1 in turn.)

The overall sensitivity results indicated that the index variables with high sensitivities were the water area index, water resource development and utilization rate, and per capita GDP, which reached 73.13%, 38.88% and 27.10%, respectively. All of them belong to the water resources subsystem and were significantly larger than the index variables of other systems. Therefore, special attention should be given to the water resources subsystem with a high sensitivity index in the evaluation of the water resources carrying capacity.

## 4. Conclusions

This paper used the fuzzy comprehensive evaluation model to evaluate the carrying capacity of water resources under the weight comparison method. The evaluation results indicated that the 11 townships in Zanhuang County had good bearings. At the same time, the sensitivity analysis method based on the index weight was used to identify the index results with the greatest influence. The evaluation results showed that the sensitivities of the water area index, water resource development and utilization ratio, per capita GDP, water consumption per unit regional GDP and domestic water quota were 73.13%, 38.88%, 27.72%, 27.10% and 15.03%, respectively. The analysis of the attribution results of the water resources carrying capacity subsystem indicated that the water resources carrying capacity of the study area was greatly affected by the regional water resources endowment and socioeconomic statuses.

The case study proved that there is a certain degree of uncertainty in the evaluation of water resources carrying capacity. The main sources of uncertainty are the uncertainty of index assignment and the uncertainty of weight. Among them, the uncertainty caused by water area index is the largest, and the uncertainty caused by fertilizer intensity index is the smallest. The uncertainty of the weight can be studied by the degree of correlation contribution. The degree of correlation contribution of the evaluation value obtained by the analytic hierarchy process is 80%, which is much higher than the 40% obtained by the entropy method. The final evaluation result obtained from the analytic hierarchy process is more in line with the actual state of good carrying capacity in Zanhuang County. Therefore, strengthening the control of the data quality of high-uncertainty indicators can help reduce errors in the evaluation of water resources carrying capacity.

Compared with previous research, the present study provides a good example of the uncertainty of water resources carrying capacity evaluation from the perspective of index sensitivity analysis and weight screening. The result is of great significance to the selection of water resources carrying capacity evaluation indicators and the practical application of evaluation results, which has great value in expanding the application of evaluation models and the development and utilization of local water resources in the future.

**Supplementary Materials:** The following are available online at https://www.mdpi.com/article/10.3390/w13202804/s1, Table S1: Evaluation criteria of 1–9 scale method, Table S2: The judgment matrix of four subsystems.

**Author Contributions:** Y.G., methodology, data analysis, software; J.W., conceptualization, formal analysis, supervision; D.Z., methodology, data provision; R.J., software, data curation; H.Y., data curation. All authors have read and agreed to the published version of the manuscript.

**Funding:** This study was funded by the National Natural Science Foundation of China (Grant No.: 41807344).

**Institutional Review Board Statement:** Not applicable.

**Informed Consent Statement:** Not applicable.

**Data Availability Statement:** Data available upon request due to privacy and ethical restrictions.

**Acknowledgments:** The authors would like to express their gratitude to the editors and anonymous experts for their constructive comments.

**Conflicts of Interest:** The authors declare no conflict of interest.

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
