# Peer review of "Uncertain Analysis of Fuzzy Evaluation Model for Water Resources Carrying Capacity: A Case Study in Zanhuang County, North China Plain"

_water, doi:10.3390/w13202804_

Round 1

Reviewer 1 Report

The authors are complemented for a very well written and referenced paper, clear outline of objective, methodology, analysis, and evaluation. This work builds-on and compliments prior work and should be a valuable reference for practical planning and future research.

Only two editorials:

Line 179-180: The sentence “The other way around…” is not clear and recommend rewording.

Line 285: The sentence “Figure 3 is the…” seems out of context or redundant. 

Reviewer 2 Report

In the submitted manuscript, Ge et al., tried to evaluate the carrying capacity of water resources using a comprehensive fuzzy evaluation model in Zanhuang County, China. Moreover, both the analytic hierarchy process and entropy method are used to determine the weights under the fuzzy evaluation model. I am not confident about the overall quality of this work rather I feel this work seems to be a good report but not a research article. I cannot perfectly understand the originality of this work. What is the novelty, main contributions, significance? Model assessment using Fuzzy analysis and other integrated models were investigated comprehensively in the literature. Similarly analytic hierarchy process and entropy methods. Thus, I cannot find this work has interest to readers. In addition, the statistics of some data provided in the main text

Reviewer 3 Report

The reviewer is comfortable with the use of fuzzy methods in water resources research. These methods, because of their fuzziness, can provide deeper insights into various factors affecting water resources. However, the main impression from reading the paper is that due to the description, it is not possible to replicate the method used and the calculations performed. The paper is not clear from the methodological point of view.

Despite the fact that index types are used, it is not clear how real raw data from the field is transferred to these indexes. Further, it is also not clear how the classification standards were defined. It is stated that classes V1 to V5 are defined by expert judgement, but why are these classes non-linear in some cases, but linearly distributed in others? This needs to be justified. Reviewer has also expressed doubts about the criterion level. Some of the index levels are related but positioned in different criterion levels (e.g. water consumption per unit of GDP in A with per capita GDP and domestic water quota in D).

Define somewhere the variable »Water Resources Carrying Capacity«. It is not a variable which can be understood per se. Some of Index types have very strange terminology (e.g Ecological base flow guarantee rate; probably Minimum Required Water Flow Rate, etc.)

Maybe methods are clear and understandable for authors, but description and explanation of them is very »fuzzy«. Avoid describing relations with words that can be explained with equations. First describe mathematical background and then algorithms.

Line 287. When you look at the correlation coefficient, you need to test whether it is significantly different from zero (null hypothesis r=0). If it is not significantly different from zero, it means that you must understand it as r=0 and consequently there is no correlation between the variables. Since the value r=0.0241 in your case is very low, this coefficient is indeed zero and there is no relationship between the methods. What does this mean? Are they equally applicable?

Reviewer fairly doubt that shares can be reliable calculated to two digits (e.g see line 409).

Figure 4. Meaning of orange colors on maps?

Figure 6 and 7. What is the meaning of abbreviations?

The paper has 35 references, but all but one are from China. The reviewer is aware that scholarly work there has increased greatly recently, but nevertheless such references may be considered local. Some basic and initial references have also been published elsewhere and do not relate only to China. Apply them.

Water is international journal intended to international public. Exclude words and sentences using »... in foreign countries ...« (e.g line 42) and »The domestic research ...« (e.g. line 47).

The reviewer is not a native speaker and therefore not qualified to judge the language. However, he feels that some sentences are too long and contain too many words that are repeated over and over. Sometimes he feels that the order of words is not correct and therefore some sentences give the impression that they are not closed. There are also not a few typos (e.g. in punctuation - comma, full stop, etc.). Some sentences are really inventions (eg. lines 275-276  “This result also verifies that the entropy weight method is based on scattered data calculations and conforms to the laws of mathematics.”). Reviewers suggest that the work should be proofread by a native speaker or someone with a good command of English.

Round 2

Reviewer 2 Report

In the revised manuscript, authors tried to answer most of my questions but it was quite difficult to track my answers in the submitted cover letter by authors. In particular, Q5, Q6, and Q7 were not clearly shown in the main text of the revised manuscript. I encourage that all authors confirm the revised submission, especially the first author should ensure professional preparation of all documents.  

Minor:

  1. It is still not easy to understand how an indicator can be considered as a direct of indirect.
  2. Table S1 was included in the Supplementary materials but was not mentioned in the main revised text.

Reviewer 3 Report

I would like to thank the authors who responded constructively to my comments and suggestions.

I can agree with the authors that there are (probably) different understandings of what "water carrying capacity" is. However, what I missed is the definition of how the authors understand this term. In my opinion, it is not defined in a way that the reader can understand how the authors understand it. In lines 53-56, there is an implicit definition that I do not think is precise enough. If possible, please define the term as it should be defined, e.g. "Water carrying capacity is (defined as) ...". Why don’t include tables in Supplementary material into the paper. They are not so extensive that they can’t be included in the paper. It will improve readability of it.

Figure 3. In the cover letter you written the response: “Orange on the map means that the depth of orange is used to reflect the final evaluation value of each region. The darker the color, the higher the evaluation value, and the lighter the color, the smaller the evaluation value.”. I can't find this explanation in the paper. Please add this explanation.

In line 42 there is still »... in foreign countries ...«. There is nothing wrong with describing something for your colleagues in China, but for the international audience it is better to use a neutral description. This comment is not crucial to understanding the article, but is meant to suggest a more universal approach.

If possible include sources of data (e.g. The Shijiazhuang Water Resources Bulletin, Statistical Yearbook of Zanhuang, etc.) into the list of References. If internet links are existing, add them.

Before finishing paper read carefully paper again, in some places “space” between words is missing or “spaces” are too long.
